# Community Engagement Practices at Research Centers in U.S. Minority Institutions: Priority Populations and Innovative Approaches to Advancing Health Disparities Research

**DOI:** 10.3390/ijerph18126675

**Published:** 2021-06-21

**Authors:** Tabia Henry Akintobi, Payam Sheikhattari, Emma Shaffer, Christina L. Evans, Kathryn L. Braun, Angela U. Sy, Bibiana Mancera, Adriana Campa, Stephania T. Miller, Daniel Sarpong, Rhonda Holliday, Julio Jimenez-Chavez, Shafiq Khan, Cimona Hinton, Kimberly Sellars-Bates, Veronica Ajewole, Nicolette I. Teufel-Shone, Juliet McMullin, Sandra Suther, K. Sean Kimbro, Lorraine Taylor, Carmen M. Velez Vega, Carla Williams, George Perry, Stephan Zuchner, Melissa Marzan Rodriguez, Paul B. Tchounwou

**Affiliations:** 1Prevention Research Center, Department of Community Health and Preventive Medicine, Morehouse School of Medicine, Atlanta, GA 30310, USA; chrevans@msm.edu (C.L.E.); rholliday@msm.edu (R.H.); 2Prevention Sciences Research Center, School of Community Health and Policy, Morgan State University, Baltimore, MD 21251, USA; payam.sheikhattari@morgan.edu (P.S.); emma.shaffer@morgan.edu (E.S.); 3Department of Public Health Sciences, John A. Burns School of Medicine, Ola HAWAII, University of Hawai’i at Mānoa, Honolulu, HI 96822, USA; kbraun@hawaii.edu (K.L.B.); sya@hawaii.edu (A.U.S.); 4Border Biomedical Research Center, College of Health Sciences, University of Texas at El Paso, El Paso, TX 79968, USA; barias@utep.edu; 5Department of Dietetics and Nutrition, Robert Stempel College of Public Health and Social Work, Florida International University, Miami, FL 33199, USA; campaa@fiu.edu; 6Department of Surgery, Meharry Medical College, Nashville, TN 37208, USA; smiller@mmc.edu; 7Department of Biostatistics, Xavier University, Cagayan de Oro 9000, Philippines; dsarpong@xula.edu; 8Department of Psychiatry and Human Behavior, Ponce School of Medicine and Health Sciences, Ponce, PR 00716, USA; jcjimenez@psm.edu; 9Department of Biological Sciences, Clark Atlanta University, Atlanta, GA 30314, USA; skhan@cau.edu (S.K.); chinton@cau.edu (C.H.); kbates@cau.edu (K.S.-B.); 10College of Pharmacy and Health Sciences, Texas Southern University, Houston, TX 77004, USA; veronica.ajewole@tsu.edu; 11Center for Health Equity Research, Northern Arizona University, Flagstaff, AZ 86011, USA; nicky.teufel@nau.edu; 12Department of Anthropology, University of California, Riverside, CA 92521, USA; julietm@ucr.edu; 13College of Pharmacy and Pharmaceutical Sciences, Florida Agricultural and Mechanical University, Tallahassee, FL 32307, USA; sandra.suther@famu.edu; 14Biological and Biomedical Sciences, North Carolina Central University, Durham, NC 27707, USA; kkimbro@nccu.edu (K.S.K.); lorraine.taylor@nccu.edu (L.T.); 15Center for Collaborative Research in Health Disparities, University of Puerto Rico Medical Sciences Campus, San Juan, PR 00921, USA; carmen.velez2@upr.edu; 16College of Medicine, Howard University, Washington, DC 20059, USA; cdwilliams@howard.edu; 17College of Sciences, University of Texas at San Antonio, San Antonio, TX 78249, USA; george.perry@utsa.edu; 18John P. Hussman Institute for Human Genomics, University of Miami, Coral Gables, FL 33146, USA; szuchner@med.miami.edu; 19Public Health Program, Ponce Health Sciences University, Ponce, PR 00716, USA; mmarzan@psm.edu; 20Department of Biology, College of Science, Engineering and Technology, Jackson State University, Jackson, MS 39217, USA; paul.b.tchounwou@jsums.edu

**Keywords:** community-engaged research, translation, best practices, research centers in minority institutions, lessons learned

## Abstract

This paper details U.S. Research Centers in Minority Institutions (RCMI) Community Engagement Cores (CECs): (1) unique and cross-cutting components, focus areas, specific aims, and target populations; and (2) approaches utilized to build or sustain trust towards community participation in research. A mixed-method data collection approach was employed for this cross-sectional study of current or previously funded RCMIs. A total of 18 of the 25 institutions spanning 13 U.S. states and territories participated. CEC specific aims were to support community engaged research (94%); to translate and disseminate research findings (88%); to develop partnerships (82%); and to build capacity around community research (71%). Four open-ended questions, qualitative analysis, and comparison of the categories led to the emergence of two supporting themes: (1) establishing trust between the community-academic collaborators and within the community and (2) building collaborative relationships. An overarching theme, building community together through trust and meaningful collaborations, emerged from the supporting themes and subthemes. The RCMI institutions and their CECs serve as models to circumvent the historical and current challenges to research in communities disproportionately affected by health disparities. Lessons learned from these cores may help other institutions who want to build community trust in and capacities for research that addresses community-related health concerns.

## 1. Introduction

### 1.1. Research Centers in Minority Institutions

The mission of the Research Centers in Minority Institutions (RCMI) program, established by Congress in 1985, is to strengthen research training, conduct, and infrastructure in minority serving colleges and universities and to develop independent investigators from underrepresented populations. These include African Americans, Hispanics, American Indians/Alaska Natives, Native Hawaiians, Pacific Islanders, and persons with disabilities. Every funded RCMI must work to: (1) enhance capacity in basic biomedical, behavioral, and/or clinical research; (2) train and support affiliated investigators to become more successful in obtaining extramural funding, especially from the National Institutes of Health (NIH), to address diseases disproportionately impacting their target populations; (3) develop new and early career investigators; (4) enhance the quality of research on minority health and health disparities; and (5) establish and sustain relationships with community-based partners. Initially, research development was focused on the biomedical sciences, but subsequently expanded to include clinical research in the mid-1990s and community-based research in 2002.

The RCMI program was initially under the administrative oversight/purview of the National Center for Research Resources. In 2017, the RCMI program was moved to the National Institute on Minority Health and Health Disparities (NIMHD). RCMIs support NIMHD’s explicit vision “to advance the science of minority health and health disparities research by enabling all investigators within the program the opportunity to engage in rigorous, mentored research experiences focused on diseases that disproportionately affect minority and other health disparity populations.” [1].

This transition also dawned a new requirement that all RCMIs include Community Engagement Cores (CECs) to support minority researchers’ efforts to meaningfully engage community stakeholders in reducing health disparities experienced by indigenous and minority groups in the US [2]. These relationships are critical for addressing racial/ethnic health and healthcare disparities for many reasons, including increased relevance and uptake of disparities research when community stakeholders are meaningfully engaged [3]. They are also fundamental to bringing to fruition the National Academy of Medicine’s (formerly Institute of Medicine) recommendation that community stakeholders should be engaged in all research phases [4]. While CECs are a recent, newly required core for RCMI programs, most of the RCMI institutions, particularly those that are also Historically Black Colleges and Universities (HBCUs), have long-standing relationships with community members and organizations, making this transition seamless and inherently central to their deeply-rooted, long standing missions or visions. These institutions are, therefore, uniquely positioned to support development and sustainability of community-academic research partnerships [5]. In many cases, these partnerships benefit from the close proximity of the RCMI institutions to the communities they serve, including communities with high proportions of racial and ethnic minorities and other populations disproportionately experiencing adverse health outcomes.

### 1.2. Community Engagement Cores

Several other NIH funded infrastructure grants have or had specific goals to reduce health disparities and/or conduct translational research through U54 funding mechanisms. This funding mechanism supports a spectrum of research and related infrastructures from basic to clinical. Activities may be multidisciplinary or biomedical. A few noteworthy U54 funding mechanisms have included community engagement components. For example, the Community Network Program Centers (CNPCs), funded through the National Cancer Institute, were required to use community-based participatory research (CBPR) approaches to reduce cancer disparities in communities [6,7].The Partnerships to Advance Cancer Health Equity Program, initiated in 2001, provides cancer-related education and awareness outreach activities and develops community partnerships with underserved communities to study cancer health disparities and their impact on racial/ethnic minorities, medically underserved, and socioeconomically disadvantaged populations [8]. The Institutional Development Award (IDeA) Program was established in 1993 to enhance biomedical research activities in states that have had historically low NIH grant funding success rates. An IDeA Clinical Translation Research Program contains a community engagement and outreach core to identify priority health issues and the concerns of communities within participating states and to develop plans for building the capacity to respond to these concerns [9]. The Clinical and Translational Science Awards (CTSA) program is a national network of medical research institutions that work to improve the translational research process and to reduce the time it takes to move research from the bench to the bedside and, ultimately, to communities. Along with providing core resources, mentoring, training, and opportunities to develop innovative approaches and technologies, CTSAs must sponsor a community engagement program [10,11].

These research infrastructure programs implement specific community-engagement-related activities throughout their research networks to address their respective program goals. Activities broadly involve increasing community partnerships, engaging communities in research, building community capacity to address priority health issues including research, and implementing principles of community engagement [10,12,13]. For example, the CNPCs formed Community Advisory Boards to ensure community involvement in research and oversee the integrity of community research projects. CTSAs provide community engagement training specifically for researchers to develop their community-academic partnerships and community-engaged research [14]. Institutions may also implement activities to foster relationships and trust building with community partners, including community research training and outreach [13,14].

Facilitators of and barriers to community-engaged partnerships and research have been identified. Trust, respect and long-term commitment are interpersonal qualities found to facilitate collaboration in academic-community partnerships [10,15,16,17]. Conversely, limited investments of time, unclear roles and/or functions of partners, and mistrust are identified barriers that deter community-engaged research activities [15,17].

Additional qualitative outcome explorations have mentioned the challenge of balancing “process” over “product.” [15,17]. Accordingly, recommendations are to define community engagement beyond the common processes and outcomes. For example, Sy et al. examined the direction of engagement (community-initiated or investigator-initiated) on the level of involvement of communities in stages of the research process [18]. The CNPC network used a combination of qualitative and quantitative measures to try to document and describe their successes at improving cancer knowledge, behaviors, and resources through CBPR processes [6,19,20]. Wallerstein et al. took a more systematic approach by first developing a comprehensive logic model for CBPR, showing the flow from community context (e.g., levels of trust and capacity), to partnership processes (relationships and structures that facilitate shared funding and power), to outcomes related to partnership (increased community knowledge and skills) and community resources, as well as indicators of health and social justice [20]. This team subsequently developed and tested a tool to measure outcomes of community engaged partnerships [21,22].

### 1.3. Objective

The purpose of this paper is to detail the RMCI CECs in terms of their: (1) unique and cross-cutting components (strategies and services), focus areas, specific aims, and target populations; and (2) approaches utilized to build or sustain trust towards community participation in research. Lessons learned from these cores may help other institutions who want to build community trust and capacity for research that address community-related health concerns.

## 2. Materials and Methods

A mixed-method (quantitative and qualitative) data collection approach was employed for this cross-sectional study. In October 2020, a 17-item questionnaire was disseminated via email to former and currently funded RCMI CEC representatives utilizing Google Forms. Items solicited short, open-ended answers regarding: (1) years of funding for the RCMI and CEC; (2) status of any CEC advisory committee; (3) specific aims of their CEC; (4) services provided by their CEC; and (5) engagement strategies of their CEC. Specific aims, engagement strategies, and services were sorted into common categories and tallied.

In November 2020, a follow-up questionnaire with four open-ended items was disseminated to all respondents to glean additional insights related to community engagement and trust. Questions were: (1) What are the strategies, conscious efforts, or steps of your CEC to overcome mistrust and ensure trust in your community engagement efforts? (2) What do you think are unique best practices in working with your communities? (3) Why have those best practices worked with your community? and (4) What is an example of a success story? Data from the follow-up survey were analyzed using content analysis to identify supporting themes and their interrelationships. Analysis utilized a grounded theory approach to facilitate systematic inductive abstraction and coding of themes (Figure 1) [23]. Specifically, open coding was first used to draw out data bits (small pieces of data) needed to develop supporting themes (Figure 2). This was followed by axial coding to show the relationships between the supporting themes [23,24].

## 3. Results

### 3.1. Demographics, Components, Focus Areas, Specific Aims and Target Populations

A total of 18 of the 25 formerly or currently-funded RCMIs responded, including institutions spanning 13 states and territories (Figure 1). As shown in the map, nine (50%) of the RCMIs supported basic, clinical, and behavioral research projects, while six (33%) supported basic and behavioral only, and two supported basic and clinical only. Responding RCMIs were serving a variety of underserved populations, including African Americans, Hispanic and Latino communities, Native Hawaiians and Pacific Islanders, and other minority groups in both rural and urban settings.

Additional quantitative results are summarized in Table 1. About 41% of the RCMIs had been funded for over 20 years, while 47% had been funded less than 10 years. Three (18%) of the RCMIs had established their CECs five or more years ago. Others were newer, with 59% established within the past two years. Fifteen (88%) had formed advisory committees. The specific aims for the CEC sorted into four common areas: to support community engaged research (94%); to translate and disseminate findings (88%); to develop partnerships (82%); and to build capacity around community research (71%). To facilitate these aims, most CECs helped connect researchers with communities (77%), provided education and training around engagement (71%), and assisted with proposal development (53%). Commonly listed engagement strategies included skills building (82%), implementation of community-engaged research approaches (82%), dissemination (71%), and discussion (59%).

### 3.2. Themes from the Open-Ended Questions

Analysis of findings from the four open-ended questions led to the emergence of two supporting themes [24]. These included: (1) establishing trust between the community-academic collaborators and within the community and (2) building collaborative relationships. Each of these supporting themes, along with relevant subthemes, is described and illustrated with sample quotes from CEC leaders’ responses to the open-ended questions. An overarching theme, building community together through trust and meaningful collaborations, emerged from the supporting themes and subthemes, representing the highest level of abstraction from the data (Figure 2) [23,24].

### 3.3. Establishing Trust between the Community-Academic Collaborators and within the Community

This supporting theme emerged from recurring data, with three subthemes. They are detailed in the sections that follow.

Building trust and rapport: Critical to the success of community-academic partnerships is building trust and rapport to address mistrust. For many minority communities in the US, this mistrust has historical precedence based upon extremely negative experiences and total disregard for the protection of participants in research. Examples include, but are not limited to, the Tuskegee Study and research involving the Havasupai Indian Tribe [25,26]. Another facet of community mistrust, as an RCMI CEC leader stated, includes “researchers entering a community to serve their own interests.”

CECs have utilized various strategies to gain the trust of their partners as described by one: “While many CECs have advisory committees, ours is unique because it includes individuals who are part of a CBPR partnership that has persisted for over a decade. These individuals provide an invaluable perspective that is incorporated into our programs, communications, and strategic plans.”

An integral component of trust is transparency. It should go beyond the research itself and include, as one CEC leader expressed “how results are used and disseminated...community should get the results first.” Another CEC leader noted that building trust and rapport included the application of “practices that are respectful of the roles, expertise, and wisdom that communities have or should have in the research enterprise. These practices foster feedback and generate more appropriate options and may segue into citizen science for members of disparate communities.” This also helps by establishing respect for the culture and practices of the community and will help develop and sustain partnerships. One CEC leader succinctly stated, “Trust is gained, never claimed. To gain trust in our community engagement efforts and the research, the community must see us as transparent and caring.”

Building capacity for the researcher and community: Capacity building for researchers and the community is another important strategy that must be encouraged for building beneficial, bi-directional community collaborations. One CEC leader noted, “Always include your community consultants/advocates or any community member who makes significant contributions to your projects as co-authors on presentations and publications.” Another stated, “Encourage community members interested in research to earn graduate degrees and become researchers for their communities.”

Aside from collaborating on grant submissions for funding, it is important to recognize that many community partners have very limited financial resources. One method successfully utilized by an RCMI institution to build community capacity is to:

Provide seed funding for small projects to help the partnership develop and strengthen before jumping into a larger project with more funding. Seed funding gives partners a unique opportunity to discuss roles and responsibilities and to negotiate shared decision- making and power early on, which strengthens the partnership in the long-term.

Another CEC leader shared that members of its CEC community advisory group can “access $2500/year for their own personal development, e.g., to attend a local or national conference to present on an RCMI project, to gain research skills, and to expand their networks.” This RCMI also offered funding to community groups to host community-based research.

Empowering the community: One of the goals of community engagement should be empowerment. A CEC leader wrote about the importance of “facilitating the translation of scientific concepts and methods needed for personal and community decision-making.” An RCMI CEC leader shared this innovative approach towards community empowerment:

“We support a community-based research group which vets requests from students, instructors, and researchers who want to access the community. Instructors, researchers, and students must attend a few monthly meetings to get to know the group and then present and defend their research, explaining how they will build community capacity, share data [and] share findings, etc.”

Another noted that “A sign of successful empowerment is when community members speak out when they are feeling used or they don’t like the direction taken by a research project. This requires a high level of trust with university partners, as well as sufficient capacity with research.”

### 3.4. Building Collaborative Relationships

Each RCMI institution utilizes validated approaches and best practices to engage with their diverse communities. Three subthemes, responsibility to the community, involvement, and relationship building, supported the theme of building collaborative relationships.

Relationship building: Relationship building is the premise and nature of community engagement. The interactions between community members and academics participating in community-engaged research require thoughtful and meaningful, bi-directional interaction and reciprocity, so as not to take advantage of the community. Since each institution and community differs, models may vary. One such model borrowed and adapted from academe that has worked for numerous RCMIs was well described by a CEC leader:

“We use a shared governance model and integrate community partners in leadership of the program. Community partners are co-leaders of the CEC and members of the program advisory committee that assesses progress and outcomes and makes recommendations for improvement.”

Relationships are “built upon the trust,” which is also earned by using a shared governance model, as well as by building capacity in research, as noted above. “The collaborative nature of the existing relationships will help guide additional work,” explained a CEC leader.

For community, a strong relationship with university partners increases their access to people and/or groups in positions of authority and power, known as “social leveraging.” Similarly, university partners’ linkages to groups outside the community, such as government institutions, policy makers, businesses, and funders, can provide key resources to develop their capacities, relationship and pathways to community credibility and trustworthiness [27,28]. As a CEC leader stated, “best practices in community engagement are those that produce interactions that develop into networks and resources that benefit the community and the research.”

Long-term involvement: Involvement is important, as summarized by one CEC leader: “Before anything, there must be a recognition of the importance of involving community perspectives in research and how investigators must look to communities for questions and answers on the research that involves them.” However, this is not enough. Many researchers only remain committed to their communities for a short period of time, for example, during a research project. However, in order to establish meaningful and fruitful community-academic relationships, a commitment, over time, is critical. Ideally, involvement in the community entails being involved “in community and cultural activities, and economic growth." As a CEC leader stated, through long-term involvement, researchers learn to “appreciate the community, their contributions, and experiences. This knowledge can add value to and strengthen RCMI program and research processes.”

Responsibility to the community: Researchers have a tremendous responsibility to the community to understand and acknowledge that the community has the right to express their needs. Our first responsibility is to listen to the community. One CEC leader stated, “We have learned that our research must respond to the communities’ imperatives, their perceived needs, from their point of view.”

Another responsibility relates to implementing authentic processes to gain consent and share research findings. As one CEC leader said, “We are obligated to inform health disparities research participants about the ’right to know’. It is an imperative of ethical research to inform participants not only of the procedures that involve their participation in the study protocol, but also to respect the right to know about the results of the research.” Related to this is the responsibility to discuss data ownership issues. A CEC leader expressed “A goal of our work as researchers is to support community ownership of data and the right to hear research findings first.”

Finally, community engaged researchers should take responsibility to help the community use research findings to support structural changes in the community. For example, a CEC leader reported on a project that resulted in “treating contaminated water, which affected the health of a community.” Another CEC leader noted the responsibility to give back to the community, saying: “Our PIs have created strong links with the school system and offered a series of lectures and informative materials for the county school system, which is a long-term partner.”

The word cloud in Figure 3 captures many of the concepts derived from our findings that are crucial to the work we do collaboratively within our communities. The most frequently mentioned words in the qualitative findings appear largest in the cloud, and include community, partnership, meeting, language, feedback, support, share, and finding. The prominence of these words reflects respondents’ views that community and partnership must be central to all community engaged endeavors, that meetings (which take time) are essential for sharing and feedback, and that the community must be supported and see that researchers hear their voices.

## 4. Discussion

The purpose of this paper was to detail RMCI CECs’ (1) unique and cross-cutting components (strategies and services), focus areas, specific aims, and target populations; and (2) approaches employed to build or sustain trust towards community participation in research. Housed in institutions of higher education, RMCIs can train, serve, and engage underrepresented populations in health disparities research. Related response strategies represent important opportunities for network-wide leadership in advancing translational research that results in community transformation.

Establishing community research governance can be challenging when: (1) academicians have not previously been guided by an understanding a community’s ecology, (2) community members have not led discussions regarding their health priorities; or (3) academic and neighborhood leaders have not historically worked together as a single body with established rules to guide roles and operations [20,29,30]. Further, communities that have experienced exploitation in research or other social systems (economic, political, racial/ethnic) require priority places at the research development and implementation table. Community members must be offered leadership roles, co-identify health concerns, and co-develop actionable responses. With the requirement of CECs, RCMI institutions have made this a priority. In response to the well-established aforementioned barriers and facilitators to community engaged partnership and research, several CECs detailed the role of community governance and leadership as central to relationship building and sustainability. They reflect a deep investment in collaboration, beyond that of a single research project. This approach requires an investment of time, over years, and attention to reversing traditional imbalances in power between academics and communities inhumanely and/or superficially engaged in research in the past.

Evaluating the success of community-engaged partnerships requires attention to both the “process” and “product” of the research [14,16]. Accordingly, in addition to publications, presentations, and grants for researchers, evaluations need to measure the extent to which community capacity is built and how health resources are made more available to sustain programs that can improve community health. Success may be dependent on the direction of engagement (community-initiated or investigator-initiated) or the level of involvement of communities in different stages of the research process [19]. In addition, identifying specific factors of successful community engagement are useful for standardizing future research [14].

To ameliorate these longstanding concerns, CECs discussed the imperative of long-term responsibility to the community through addressing their community priorities, alongside or independent of a Center’s research projects. This is a central tenet of social exchange that is bi-directional, thereby increasing the likelihood of sustained collaboration, trust and success. As indicated by some CECs, an academic partner can build a partnership with a community through seed funding and resources for professional skill building development. CECs can provide services that may include education or capacity building workshops to strengthen neighborhood-driven initiatives through topics like evaluation, leadership, health promotion, and disease prevention, which are important to community partners. Training and workshops led by community members are invaluable for increasing community knowledge and cultural humility among academicians. Traditional academic faculty often diminish these activities because they are time-consuming and appear to be unrelated to academic success. However, these activities are critical to the mission of reducing health disparities in the communities they serve [29].

The RCMI institutions and their CECs serve as models to circumvent the historical and current challenges to research in communities disproportionately affected by health disparities. While the benefits of community engagement have been well documented [8,11,12,13,14,15]. there are challenges in building and maintaining research partnerships between community and academic researchers. Academic researchers, who have been traditionally trained to conduct “independent” or “investigator-initiated” research, often make unilateral decisions and, consequently, have poor participatory communication skills (e.g., making decisions without input, infrequent communication). There are also few incentives for community engagement in academia, and, historically, community engaged research efforts have not been duly recognized as part of the tenure and promotion reward system. As a result, limited time is typically spent developing and committing to the partnership, even among those who may want to establish them [29,30]. Even when research is carried out in community settings, there may be little if any input from or engagement of community members, beyond requests to participate in a study or clinical trial. Plans for sustaining the intervention may be unsuccessful because the flow of information back to communities is less of a priority, as is the translation of this knowledge [29,31,32,33]. Community members and leaders have less familiarity with research processes and requirements related to institutional review boards and research designs. Moreover, research mistrust due to personal negative experiences and/or the national and global residue associated with the historical mistreatment of minority populations by researchers are understandably still pervasive [34,35,36]. These trends are being slowly reversed in academic institutions whose CECs are integrated as institutional resources, rather than siloed in discrete research projects. This institutional integration of CECs democratizes access to resources designed to facilitate support in the community partnership development, research implementation, and dissemination and also positions community engagement as a valued pillar of the surrounding neighborhoods.

### 4.1. Strengths of the RCMI CEC Network

The strength of the RCMI CECs is their historic affiliation and service within the underserved and vulnerable communities in which their respective institutions are located, coupled with the trust that has developed over time between their academicians and the community. The CECs have developed and nurtured bi-directional collaborative relationships with their respective communities for decades, and in the case of HBCUs, over a century. Since many HBCUs, Hispanic serving, Pacific Islander- and Native American- serving institutions are RCMI grantees, they share a common goal to eliminate health disparities. The CEC network also brings together a group of diverse community engagement practitioners from across 13 states and territories with unique backgrounds, experiences, perspectives, and interdisciplinary approaches coalesced around solving health disparities. The CEC network also affords a tremendous opportunity for collaborating and sharing innovative best practices towards the elimination of health disparities. Lastly, the expertise of senior scientists in the CECs facilitates mentoring opportunities for junior scientists as they embark in health disparities research and science.

### 4.2. Strengths of Study

The strengths of the study include a mixed-methods design that incorporated a quantitative survey and qualitative open-ended questions that captured, complimented, and created a more complete picture of the best community engagement practices at RCMIs. Another strength was the purpose of the study in detailing unique strategies, services, and approaches towards building and/or sustaining trust within diverse communities to improve and increase participation in research.

### 4.3. Limitations of Study

Study limitations are also acknowledged. A major study limitation was that only 72% (18 of 25) of past of currently funded RCMI CECs participated, despite numerous attempts to solicit the participation of all. Potential reasons for not participating could have been the short timeline for data collection (several months) and the fact that data were collected during the COVID-19 pandemic and by e-mail, rather than at the recurring annual meeting of the RCMIs. While we don’t expect that there would be major changes in the main categories of themes such as building trust and meaningful collaborations, participation of additional non-RCMI minority serving institutions could lead to better understanding of types of services and engagement strategies relevant to their specific target groups. Moreover, the qualitative data (as well as the quantitative data) were collected through online forms rather than through real-time interviews. Thus, some answers were shorter than others, and some answers were not clear. In these cases, one of the authors reached out to seek expansion and clarification of responses. This approach worked well, as the majority of such requests were addressed quickly and comprehensively. However, future researchers should consider conducting in-depth interviews.

## 5. Conclusions

In conclusion, this study was designed to present best practices utilized by RCMI CECs through a mixed-methods data collection approach to detail: (1) unique and cross-cutting components, focus areas, specific aims and target populations; and (2) approaches utilized to build or sustain trust towards community participation in research. CEC approaches and lessons learned may help other institutions that want to build trust and capacities for research that addresses community health concerns and leverages community strengths to advance health equity.

## Figures and Tables

**Figure 1 ijerph-18-06675-f001:**
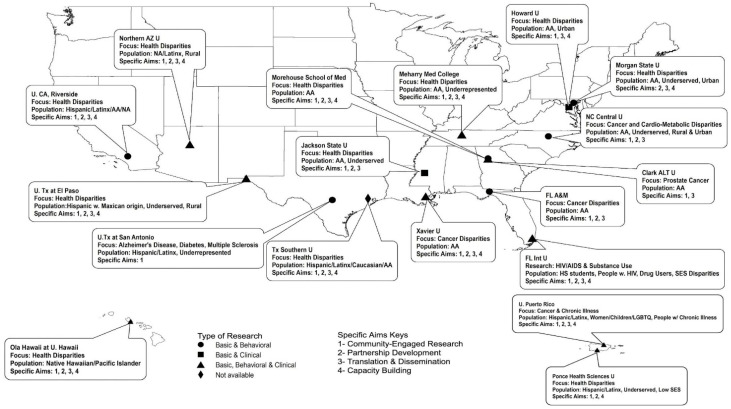
RCMI Community Engagement Components, Specific Aims & Target Populations.

**Figure 2 ijerph-18-06675-f002:**
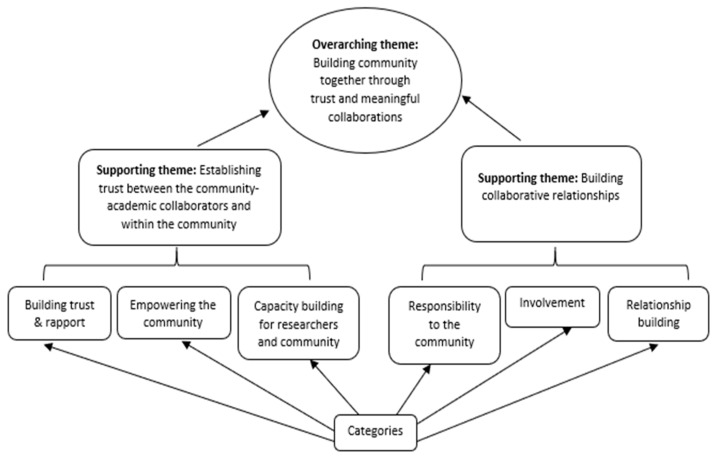
Categories, Supporting Themes and Overarching Theme.

**Figure 3 ijerph-18-06675-f003:**
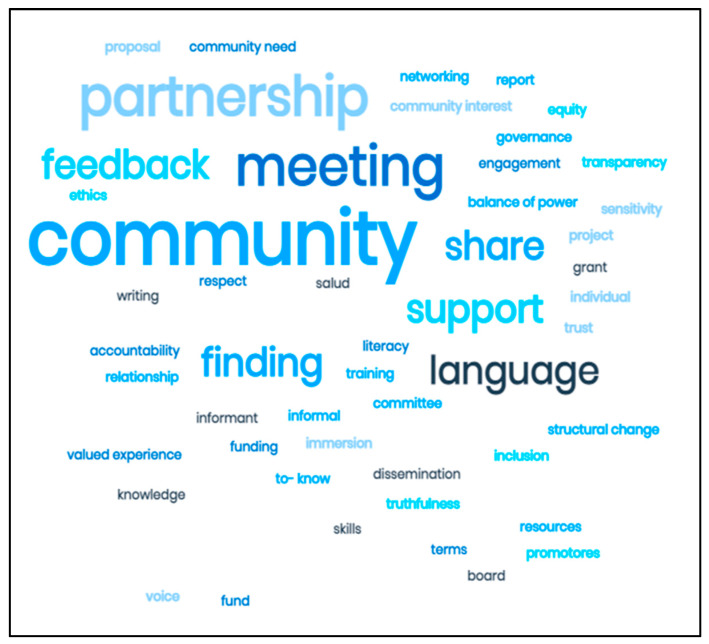
Word Cloud Associated with Community Trust.

**Table 1 ijerph-18-06675-t001:** Research Centers at Minority Institutions (RCMIs) Community Engaged Research Description.

RCMI Characteristic	Number (%)
Years RCMI Funded	
<10 Years	8 (47%)
10–20 Years	2 (12%)
20+ Years	7 (41%)
Years CEC Funded	
1–2 Years	10 (59%)
3–4 Years	4 (24%)
5+ Years	3 (18%)
Advisory Committee	
Advisory Committee Formed	15 (88%)
Advisory Committee Active	14 (82%)
Specific Aims	
Community Engaged Research	16 (94%)
Translation & Dissemination	15 (88%)
Partnership Development	14 (82%)
Capacity Building	12 (71%)
Services	
Partnership Linkage/Connection	13 (77%)
Education, Training, Tech Asst., Workshop, Seminar	12 (71%)
Proposal Development, Research Support/Funding	9 (53%)
Dissemination Events/Support	8 (47%)
Consultation	6 (35%)
Engagement Strategies	
Skill Building & Learning	14 (82%)
Implementation	14 (82%)
Dissemination Efforts & Support	12 (71%)
Discussion & Dialogue	10 (59%)
Communications	8 (47%)

## Data Availability

Raw data are available from the first author.

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
