# Peer review of "Community Engagement Practices at Research Centers in U.S. Minority Institutions: Priority Populations and Innovative Approaches to Advancing Health Disparities Research"

_ijerph, 2021, doi:10.3390/ijerph18126675_

Round 1

Reviewer 1 Report

The paper's topic is very important and would be of interest to a wide range of scholars since "community engagement" is currently a hot issue  in academia in the U.S.  The paper's relevance to "minority institutions," most notably, Historically Black Colleges and Universities (HBCUs), is also a strong point that will generate broad interest in the study's findings among scholars and practitioners.

The paper's main strengths derive from the results presented in Figures 1, 2, and 3.  These three presentations (which rest on data collected and analyzed through a fairly sound research methodology) make the paper a very compelling candidate for publication.  My only suggestion about these figures is that Figure 2 might be displayed vertically rather than horizontally, with the "Overarching theme" being placed at the top and the six "categories" forming the base.

The major problem is that the manuscript's writing is in pretty bad shape.  Throughout the paper there are numerous sentence fragments, missing words, grammatical errors, and other infelicities that sorely detract from the overall impact of the study.  There are too many problems for me to list one-by-one, but a good (or bad) example is line 414: "Naive" should be "Native."  The authors should enlist the services of a professional copyeditor and put the manuscript through two or three rounds of good copyediting.  Another problem is that many ideas are unnecessarily repeated throughout the paper and thus there are serious redundancies that are distracting.  For instance, lines 445-463 repeat verbatim earlier parts of the manuscript without justification.  Overall, the authors should strive for a shorter, more concise paper.

In conclusion, the paper's content is valuable and has the potential for a noteworthy contribution to our understanding of community engagement by minority-serving institutions of higher education.  Hence, the paper is potentially publishable.  However, in its present form, the bad writing undermines the quality of the presentation and weakens the credibility of the authors.

Author Response

Please find response to Reviewer 1 in the attached Word document

Reviewer 2 Report

The present manuscript entitled “Community Engagement Practices at Research Centers in Minority Institutions: Priority Populations and Innovative Approaches to Advancing Health Disparities Research” is designed to present best practices utilized by Research Centers in Minority Institutions Community Engagement Cores (RCMI-CECs) in a systematic approach through mixed-methods data collection that detailed 1) unique and cross-cutting components, focus areas, specific aims and target populations and 2) approaches utilized to build or sustain trust towards community participation in research. A total of 18 out of 25 institutions participated in the current study. The findings of this study may help other institutions who want to build trust and capacities for research that addresses community-related health concerns and leverages community strengths with health disparity populations towards advancing health equity. The authors found that the specific aims for the CEC sorted into four common areas: to support community engaged research (94%); to translate and disseminate findings (88%); to develop partnerships (82%), and to build capacity around community research (71%). To facilitate these aims, most CECs helped connect researchers with communities (77%), provided education and training around engagement (71%), and assisted with proposal development (53%). Four open-ended questions, qualitative analysis and comparison of the categories led to the emergence of two supporting themes: 1) Establishing trust between the community-academic collaborators and within the community and 2) Building collaborative relationships.

It is a well-designed and well-analyzed study. I have the following comments.

Typo in line 25. I think it is RCMI and not RMCI.

An unwanted period in the middle of line 68.

Typo in line 139.

An extra space in line 338.

I think the font size is not same throughout the manuscript. It looks like line 27 has a higher font size than others. I request the authors to kindly cross check the aesthetics of the manuscript. It gives a bad impression.

As 18 out of 25 RCMI institutions participated in the present study, how do the authors address the issue that it’s not a complete representation of RCMI CEC institutions?

If HBCU, Hispanics, American Indians/Alaska Natives, Native Hawaiian and other Pacific Islanders were to be included in the current study, how would that have been impacted the outcome of the study? Can authors shed some light on this?

Reviewer 3 Report

This paper discusses an important, pertinent, and timely topic in the community engagement practices of the Research Centers in Minority Institutions (RCMI) program.  Authors illustrative innovative ways to address health disparities and advance research by describing how the RCMI “established by Congress in 1985, is to strengthen research, training, conduct and infrastructure in minority serving colleges and universities and to develop independent investigators from underrepresented populations”.

Line 68 – Check for a stray period in the statement ‘Engagement Cores (CECs) .to’. 

Line 86 – An explanation of what U54 funding is would be helpful to the lay reader.

Line 138 – The objective of the paper should be introduced earlier in the paper to help set the stage for the paper.  There is a good overview on Research Centers in Minority Institutions in the introduction.  Perhaps the objective can be moved to after the first section in the introduction.

Overall, this is a comprehensive, insightful, and thorough investigation.  Attending to a few clarifying items may help to improve the paper.

Round 2

Reviewer 1 Report

There are still many typos and other editorial problems.

Lines 5-24  The list of authors doesn't match the email address list

Line 33 Italicize 1)

Line 55 change (3) to 3)

Line 83 say "populations disproportionately"

Lines 111-112 clarify "engaging health disparities communities" Do you mean "engaging communities' health disparities"?

Line 179 say CECs not CEC

Line 194 Italicize 1)

Line 252 a period (not a comma) should be after "empowerment"

Line 302 a period (not a comma) should be after "growth"

Line 305 delete "we"

Lines 337-338 the semicolon should come after "participation" not "and"

Line 380 a semicolon (not a comma) should come between "success" and "however"

Line 398 delete comma after "process"

Line 402 "These" is erroneously in superscript

Line 406 clarify "external positions community engagement" Is a word missing in the sentence?

Line 408 CE should be CEC

Line 417 delete "bring"

Author Response

Please find attached my responses to the reviewers
